# Development and Characterization of Bioactive Polypropylene Films for Food Packaging Applications

**DOI:** 10.3390/polym13203478

**Published:** 2021-10-11

**Authors:** Adrian Krzysztof Antosik, Urszula Kowalska, Magdalena Stobińska, Paulina Dzięcioł, Magdalena Pieczykolan, Katarzyna Kozłowska, Artur Bartkowiak

**Affiliations:** Center of Bioimmobilisation and Innovative Packaging Materials, Faculty of Food Sciences and Fisheries, West Pomeranian University of Technology, Szczecin, Klemensa Janickiego 35, 71-270 Szczecin, Poland; urszula.kowalska@zut.edu.pl (U.K.); mstobinska@zut.edu.pl (M.S.); paulina.dzieciol@zut.edu.pl (P.D.); magdalena.pieczykolan@zut.edu.pl (M.P.); katarzyna.kozlowska@zut.edu.pl (K.K.); artur.bartkowiak@zut.edu.pl (A.B.)

**Keywords:** bioactive substances, polyolefin-based films, packaging, processing technologies

## Abstract

Bioactive polypropylene (PP) films with active agents) presence for food packaging application have been prepared and characterized. The novel modified PP films were obtained via PP/additives systems regranulation and cast extrusion. The influence of two types of plasticizers (natural agents as well as commercial synthetic product) and bioactive additives on final features, e.g., mechanical properties, was evaluated. Moreover, the biocidal activity of the films was determined. Due to their functional properties, they are developed as additives to packaging plastic materials such as polyolefins. The study results presented in our work may indirectly contribute to environmental protection by reducing food waste. The aim of the work was to obtain innovative, functional packaging materials with an ability to prolong the shelf life of food products. The best antimicrobial properties were observed for the sample based on 5 wt.% oregano oil (OO) and 5 wt.% cedar oil (OC) in PP matrix. A microbial test revealed that the system causes total reduction in the following microorganisms: *B. subtilis*, *E. coli*, *S. aureus*, *P. putida*, *C. albicans*, *A. alternata*, *F. oxysporum.*

## 1. Introduction

Recently, consumers have become more aware of health issues and eat more nutritious food. They consume lot of fresh fruits, vegetables and they are willing to pay a higher price for better quality products. Thus, the demand for local and domestic fruits has greatly increased throughout all seasons. There are many obstacles to the delivery of fresh products, e.g., limition of the ability to deliver fresh, high quality products such as the seasonality of crops and the need to transport off-season products from various geographic areas. Long-distance transport can affect, e.g., by physical injury, the appearance of microbial flora, water loss, and changes in storage temperature. In order to extend freshness, more emphasis is placed on packaging methods. Studies presented in the literature has shown that proper packaging can extend the shelf life of products by up to 100% of the storage time. They constitute a physical barrier between the external environment and the packed products. From a technological point of view, they form a separate part of the space that protects the product, e.g., food, from damage or faster deterioration. The modifications of the film with bioactive agents presented in the paper allow us to extend the life of the food product by stopping the development of microorganisms in its packaging [1,2,3,4].

In addition to EU standard requirements for the materials used in packaging focus on usage of biodegradable or easily recyclable materials. The packaging materials should be suitable for contact with food products. For this reason, products such as fruit and vegetables are often packaged in monopolymer-based heat sealed bags or stretch wrapped trays. This lead to extend the freshness of the products, formation of independent microclimate in the space separated by the packaging. Therefore, to sustain the product freshness, more manufacturers add bioactive agents into packaging materials. It is common practice to cover the inner side of the film with a layer containing these agents (such as varnish) or to introduce them into the inserts, while it is rare to introduce bioactive agents in situ to the packaging film. In this way, the active compounds contained in the packages are released into the environment, separated by the package and limits microorganism growth on the surface of fruit and vegetables. Bioactive agents are more accepted additives than commercial antimicrobial substances such as antibiotics which have similar activity [5,6,7,8,9,10].

Tomatoes belong to the most popular group of crop plants. They contain a great deal of vitamins (e.g., K, PP and B group), macro- and microelements necessary for the proper functioning of human body. Due to their valuable properties, the stage of storing tomatoes is very important, because their speed of ripening can be manipulated by applying certain conditions. According to the literature, too low storage temperature (blow 5 °C) may lead to a loss of sensory quality, or cell or tissue damage to the tomatoes. The most optimal storage conditions for harvested tomatoes are temperature range between 18 and 21 °C, but the usual cold chain is ca. 12 °C. Depending on the storage conditions by consumers, tomatoes may have different freshness times. Tomatoes stored at higher temperatures, e.g., at room temperature, lose their freshness earlier than at lower temperatures, because degradation of tomato tissues are much faster. To extend its shelf life, both at high and low temperatures, commercial polypropylene (PP) or polyethylene (PE) are modified with different natural additives with antimicrobial properties and they are approved for food as safe packaging materials [11,12,13,14].

In our study, we examined the antimicrobial and antifungal activity of essential oils (EO). They form a diverse group of chemical agents that include, e.g., aldehydes, alcohols, esters, and phenols. Essential oils with potential antimicrobial properties were selected from the literature. The tested substance was oregano oil. This oil exhibits antioxidant, anti-fungal, anti-inflammatory properties, and is approved for food contact. Rosemary extract has been used in food preservation to prevent microbial contamination and oxidation. As a preservative, it replaces synthetic antioxidants in foods. Rosemary extract is commonly used in medicine. In our study, we also used methylparaben as a substance to enhance antibacterial properties. Parabens are applied as anti-microbials in foods, pharmaceutics and cosmetics. In the European Union, methylparaben is limited in application. In the United States, this substance is declared by the FDA to be “generally recognized as safe” (GRAS). Green tea containing, e.g., catechins, are polyphenolic compounds. Green tea extracts possess anti-tumor, anti-inflammatory, anti-oxidative, anti-microbial, anticarcinogenic properties [15,16,17,18,19,20].

Polypropylene (PP) is one of the two most commonly used plastics, next to polyethylene. PP is characterized by very good chemical resistance, high thermal expansion, good weldability, high tensile strength, and an operating temperature from −20 °C to +100 °C (as a product). It is aseptic and physiologically inert, and does not absorb moisture. The tests showed greater stiffness and hardness than PE. PP was selected for the tests by its properties described in the article and for modifications with active agents and plasticizers. PP is a stable material and widely used in packaging as well as being easy to recycle. Currently, PP is the one of the most frequently chosen plastics for the foil production. The stability of the processing (both in regranulation and in cast extrusion) makes it a desirable polymer matrix for modification [21,22,23,24].

This work uses a bioactive product as an additive for modified PP films used in the packaging industry. In addition, plasticizers will be added, for supporting a migration of bioactive agents to an environment. The objectives of this study were to develop an active PP film containing bioactive agents by an extrusion (mixing of PP and additives), cast film extrusion method and to characterize the films physical properties. This study also evaluated the antimicrobial and biostatic properties of the developed films. Finally, this investigation examined the influence of selected films on the storage stability of tomatoes at simulated conditions. Similar research has already been carried out, where films with antimicrobial properties based on PLA with oregano oil were obtained [25]. We expected a similar or better effect after introducing oregano oil with the plasticizer to the PP matrix.

## 2. Experimental

### 2.1. Materials

Commercial polypropylene was used (acronym: HP515M, Melt Flow Rate (230 °C/2.16 kg): 9.0 g/10 min, Density: 0.90 g/cm³, Vicat Softening Temperature (A/50 N): 153 °C), which was product of BasellOrlen (Poland). As plasticizers, Atmer 121 (A121), product of Croda (UK), and cedar oil (CO), isolated form pine nuts (*Pinus sibricae),* product of Ekamedica (Poland), were applied. Rosemary extract (RE), product of Exeller (Belgium), oregano oil (OO), product of Hepatica (Turkey), methylparaben (MP), product of Chmes (Vietnam), and green tea extract (GTE), product of Exeller (Poland), were used as bioactive agents.

### 2.2. Preparation of PP-Based Films–Regranulation and Cast

#### 2.2.1. Regranulation

The regranulation process was performed using twin screw extruder (10 heating zones, Labtech Engineering, Thailand). PP pellets were extruded with plasticizers and various bioactive agents (as powders, pastes and liquids) in order to obtain uniformly blended pellets. The following during the tests were applied 170–190 °C and 170–185 °C processing parameters for mono-modified systems and mix-modified systems, respectively (Appendix A).

#### 2.2.2. Cast

The final native and modified PP pellets were extruded through a flat die using Multilayer Chill-Roll Cast Film Extrusion Line Type LCR-300 Co-Ex (Labtech Engineering, Thailand) in a single-layer mode in order to obtain a films with a thickness of 50 to 150 µm. The following temperature profiles during the tests were as follows: 175–210 °C, 175–195 °C and 160–185 °C temperature processing parameters for native PP, mono-modified pellets and mix-modified pellets, respectively (Appendix A).

#### 2.2.3. Experimental Design

The experiment was designed in the four stages (Appendix A). In the first stage, the influence of individual components on the properties of the obtained films was determined and individual correlations (two-component films) were investigated. The research confirmed the synergism between the bioactive agent and the plasticizer (the plasticizer facilitated the release of the bioactive agent, acting as a biocide by increasing the concentration of the released additives). Taking advantage of this, we attempted to increase the biocidal activity of the film while maintaining its processing and functional properties—stage 2 and 3, obtaining three-component and multi-component films, respectively. In the last, fourth stage, the best films were selected and preliminary storage tests and weldability tests were carried out. The experimental design used single-factor designs for trials in the scheme of independent groups. The statistical analysis was based on ANOVA and was supported with statistical comparative analysis (Duncan).

#### 2.2.4. Selection of Representative Strains of Bacteria

In microbiological experiments, four preselected bacterial strains, one yeast and four fungi strains were applied (the choice based on the most popular microflora occurring on surface of fresh vegetables and fruits). The microorganisms were purchased from a German Collection of Microorganisms (DSMZ): *Bacillus subtilis* DSM 1090, *Pseudomonas putida* DSM 6125 and also *Staphylococcus aureus* DSM 346 and *Escherichia coli* DSM 498, which are indicated in the antimicrobial standards ASTM E 2180-01 [26]. From Polish Collection of Microorganisms (PCM), *Candida albicans* PCM 2566. The fungi strains were from the Czech Collection of Microorganisms (CCM); *Alternaria alternata* CCM F-397, *Fusarium oxysporum* CCM F-545, *Penicillium expansum* CCM F-576 and *Aspergillus brasiliensis* CCM 8189 were obtained.

#### 2.2.5. Selection of Final Antimicrobial Substance

Oregano oil, rosemary extract, methylparaben and green tea extract have been identified as the most promising antimicrobial substances according to the authors of [27,28,29,30]. The MIC (Minimal Inhibitory Concentration) tests were carried out, as the recommended method by EUCAST (European Committee on Antimicrobial Susceptibility Testing). The results confirmed the antimicrobial activity of these substances and the selection of the appropriate concentration are presented in the results.

### 2.3. Methods

#### 2.3.1. Mechanical Properties

Mechanical tests were carried out on the Zwick/Roell Z 2.5 machine, Ulm Germany (2.5 kN head) according to ASTM D822-02. The samples were cut into strips of 15 mm in width and placed between clamps with 50 mm distance and elongated at tensile speed 100 mm/min. At least seven replicate samples were tested. The elongation at break, maximum tensile strength, Young’s modulus with standard deviations were calculated with TestXpert II software.

Melt flow index (MFI) was carried out on the RB-M Plastometer machine, Rolbatch GmbH, Germany, according to the standard tests carried out in accordance with the manufacturer’s instructions according to the standard method A and according to ISO 1133 (load 1.26 kg, temperature 230 °C).

#### 2.3.2. Thermal Characteristics

Thermal characteristics were assessed by thermogravimetric analysis (TGA) using TA Instruments Inc. model 2950 TGA unit interfaced with the TA Instruments Thermal Analyst 2100 control unit. All samples, of about ca. 10 mg, were placed in the platinum pan and analyzed with air atmosphere at 60 ml/min during the thermo-analysis process. The temperature was ramped at 5 °C/min.

#### 2.3.3. Infrared Spectroscopy

FTIR analysis was performed using the Nexus (Thermo-Nicolet, Waltham, MA, USA) technique equipped with ATR. For each sample. A total of 32 scans were taken from 4000 to 400 cm^−1^.

#### 2.3.4. Antimicrobial Properties

The antimicrobial properties of PP films with active agents were evaluated according to ASTM E 2180-18 standard. Samples of extruded PP films of 3 cm × 3 cm with active substances were prepared. The concentrations of the microorganism were standardized to 1.5 × 10^8^ cfu/mL and suspended in 100 mL of 0.85% NaCl with 0.3% agar. The final concentration of each strains was 1.5 × 10^6^ cfu/mL in agar suspension. Inoculated agar (1.0 mL) was pipetted onto each square film sample and introduced into the sterile Petri dishes. All samples were incubated in climate chamber for 24 h (bacteria and yeast), 48 h (fungi) at 30 °C with relative humidity at 90%. After incubation, the samples were introduced into the 100 mL of tryptic soy broth medium (TSB, Merck Germany) and serial dilutions of the initial inoculum were performed. Each dilution was spread into the agar medium and incubated for each strain’s reference culture temperature. All measurements were performed in duplicate from each active films. The obtained results were presented as an average value with standard deviations of three replicates. Where relevant, the data were subjected to one-way analysis of variance using ANOVA test.

## 3. Results and Discussion

As relatively high temperatures are applied in the PP thermo-processing (both regranulation and cast extrusion), which could cause an undesirable loss of the bio-additives quantity, adjusting them to the thermal resistance of the additives is necessary. In order to evaluate a potential excessive losses of bioactive agents, thermogravimetric analysis was carried out to determine their behavior at increased temperature (Figure 1). Both methylparaben as well as green tea extract showed higher thermal resistance compared to the essential oil and rosemary extract. In the case of OO and RE, a significant loss of mass occurred at a temperature of about 230–250 °C, which is above the processing temperature described in the literature [31,32]. Due to high weight loss of the rosemary extract and oregano oil samples (100 and ca. 50% of the sample weight lost, respectively) at ca. 200 °C, the processing temperature profiles were lowered for processed PP pellets modified with individual components in the range from 175 to 195 °C on the end of line (Appendix A) in order to release as little as possible of bioactive agent (similar content to introduced amount into the premix) from the polymer matrix [23,25]. However, due to the production process (extrusion and cast extrusion), these bioactive agent (e.g., oregano oil) are trapped in the polymer bulk, migrate and evaporate from the surface of the pellet/film. Loss of additives form PP can be minimized to 15–25% via lowering the temperature profiles of the extrusion and increasing the cooling rate of extrudate [25,33,34]. The initial small weight loss each of analyzed active substance at ca. was probably ascribed to the moisture. The second step presented a significant mass loss. This loss may also be attributed to the volatilization/decomposition of bioactive compounds from the substances, e.g., for rosemary extract, possibly decomposition off phenolic diterpenes, such as carnosic acid and carnosol, as well as rosmarinic acid. For green tea extract, oregano oil and rosemary extract, the third step—small mass loss—can be related to inorganic compounds [23,25,31,32,33,34].

The influence of additives on the mechanical properties of PP films was determined. The bioactive properties of the obtained new films were checked on the basis of the obtained results. Premixes with two components were prepared to investigate their joint effect on the mechanical properties and increase the bioactive potential. It was observed that the addition of the higher amount of selected substances, e.g., oregano oil and rosemary extract, caused a better flow of the polymer (the substances exhibit lubricant properties), which lowered the processing temperatures for the regranulation and cast extrusion process (Appendix A). All studied bioactive additives/plasticizers/PP systems are presented in Appendix A with notes indicating the most suitable system for further modification using the experience observed in the preparation of film with one and two additives compositions, the three-, four- and five-component additives premixes were prepared, each time testing their joint effect on the mechanical properties and increasing their bioactive potential (the effects of the selection of modifier systems are described in section later step by step in the work).

The FTIR spectra of oregano oil, cedar oil, neat PP and modified PP are presented in Figure 2.

The samples were analyzed by FTIR spectroscopy. Spectral characteristics between films containing active substance (5% oregano oil and 5% cedar oil—Figure 2) and reference film (PP) exhibit that the major ingredients are present in the PP material. No shifts of the absorption bands were observed, which confirms the weak interaction between the introduced additives and the polymer matrix, facilitating their migration, thanks to which the biobacterial effect of the film is obtained. Comparing spectra difference between them was observed. In films containing oregano oil, the appearance of peaks from the hydroxyl groups (3940 cm^−1^) was noted. Additionally, a peak with a wavelength of 1743 cm^−1^ (carbonyl group) derived from cedar oil is observed (Figure 2) [22,33].

Most of the additives (plasticizers and antimicrobials) lowered the Young’s modulus (ca. up to 20%) and increased the tensile strength in the case of GTE and OO and elongation at break (ca. 15%) (Table 1). This can be related to the better PP orientation of the film containing both bioactive additives and plasticizers that facilitated polymer processing causing, e.g., lubricating PP transportation through extruder [35,36]. During the regranulation process, the power load showed a lower value than in the case of extrusion of native PP granulate (decrease the resistance caused by an increase in the MFI of the modified re-granules Table 1, Table 2 and Table 3. The additives act as co-plasticizers and lubricants in both compounding and film cast extrusion processes, increasing melt flowing of the premixes leading to more effective processing. The film with 5% of plasticizer: Atmer121 characterized by the highest decrease in the Young’s modulus (255 MPa), and almost highest increase in elongation at break (762%). Samples with green tea extract and oregano oil at 5% and 10%, respectively, were characterized by high values of Young’s modulus and elongation at break, which can indicate the negative effect of monoterpenes presence on PP chains. Films containing plasticizing agents showed a greasy perspiration on their surface, while films with bioactive agent exhibited a specific fragrance, which indicates the migration of additives from PP material [35,36].

The minimum inhibitory concentration (MIC) of antimicrobial substances such as OO, RE, GTE and MP ranged from 2.5 to 0.02 mg/mL. The MIC of oregano oil was 0.63 mg/mL against *P. putida* and *P. expansum,* 0.31 mg/mL, against *S. aureus*, *B. subtilis. E. coli*, *C. albicans* and *F. oxysporum*, 0.08 mg/mL for *A. brasiliensis* and 0.02 mg/mL *A. alternata*. The MIC of rosemary extract was 1.25 mg/mL against *P. expansum*, 0.31 mg/mL for *S. aureus* and *F. oxysporum*, 0.16 mg/mL *P. putida*, *E. coli*, *A. brasiliensis* and 0.04 mg/mL for the most sensitive species in our test was *A. alternata*. The MIC of GTE was 2.5 mg/mL against *P. expansum*, 1.25 mg/mL for *A. brasiliensis*, 0.16 mg/mL *B. subtilis* and *E. coli*, 0.08 mg/mL *S. aureus* and *A. alternata*, 0.04 mg/mL against *P. putida* and *C. albicans*. The MIC of MP was 2.5 mg/mL against *S. aureus* and *P. putida,* 0.31 mg/mL for *P. putida, E. coli, C. albicans, P. expansum,* 0.16 g/mL against *A. alternata* and 0.02 mg/mL for *A. brasiliensis.* The inhibitory concentrations, caused similar decrease (*p* > 0.05) in the counts of all assessed groups of the microorganisms.

Single additive films with active substance have antimicrobial properties against bacteria (Appendix A), yeast and molds (Appendix A). Films containing only commercial plasticizer Atmer121 (5%) did not exhibit any biocidal properties. Films with CO (5%) had no effect against all tested bacteria but exhibited the 1-log reduction in the number of *Alternaria alternata* (the number of the cells was 5.33 × 10^3^ cfu/mL) after surface direct contact test compared to the control sample (1.78 × 10^4^ cfu/mL). Films containing GTE (5%) demonstrated antimicrobial activity against *Pseudomonas putida*, which had total growth reduction (control was 2.28 × 10^8^ cfu/mL) and 1-log reduction in growth *Bacillus subtilis* (1.33 × 10^6^ cfu/mL) compared to control (1.47 × 10^7^ cfu/mL). Films with RE ( 5%) had no effect against all bacteria strains, but inhibited growth of molds such as *A. alternata* (2.00 × 10^3^ cfu/mL) and *Fusarium oxysporum* (2.00 × 10^3^ cfu/mL) compared to the control sample (5.47 × 10^5^ cfu/mL). Foils with additive OO (5%) reduced 1-1.5 log growth of *A. alternata* (4.2 × 10^3^ cfu/mL) and *F. oxysporum* (8.9 × 10^3^ cfu/mL). OO (7.5%) reduced growth 1-1.5 log *B. subtilis* (1.23 × 10^6^ cfu/mL) and *P. putida* (1.65 × 10^7^ cfu/mL), more than 2-log reduction in growth *Candida albicans* (3.2 × 10^3^ cfu/mL) compared to the control sample (2.75 × 10^6^ cfu/mL)*, A.alternaria* (6.5 × 10^2^ cfu/mL)*, F. oxysporum* (1.87 × 10^4^ cfu/mL) and total reduction in growth *Staphylococcus aureus*. Oregano oil (10%) reduced growth *Escherichia coli* (7.55 × 10^7^ cfu/mL) compared to the control (3.68 × 10^8^ cfu/mL) and had total reduction in growth *S. aureus*, *C. albicans, A.alternaria* and *F. oxysporum*.

Most of the added additives (bioactive and plasticizing) increased tensile strength, Young’s modulus and elongation at break, similar to the case of film modified with a single additive (Table 2 and Table 3). This may be due to the better polymer chains orientation in the film with the addition of both bioactive additives and plasticizers. The processing behavior of PP with the additive mixtures is the same as in the case of PP with single additives. Almost all samples containing additives were exhibit higher values of Young’s modulus and elongation at break than neat PP films. It can be caused by monoterpens (present in essential oils) able to cut the PP chains while plasticizing the film during thermos processing. Films with both plasticizing and bioactive agents showed the greasy droplets on the surface and a more smellable compared to mono systems, that can be related to the faster migration of additives from the material [35,37].

The antimicrobial properties for 10 various compositions of double-additive extruded films with different concentration of active substances were tested and summarized for bacteria (Appendix A), yeast and molds (Appendix A). The first mixture system of RE (2.5%) with CO (2.5%) had no effect against all bacteria strains; however, it did reduce by 1-1.5-log growth of *F. oxysporum* (5.63 × 10^3^ cfu/mL) compared to the control (5.47 × 10 cfu/mL). While increasing the concentration of RE to 5% at constant amount of CO (2.5%) the reduction in *B. subtilis* (3.85 × 10^4^ cfu/mL) while control was (1.47 × 10^7^ cfu/mL), total reduction in *S. aureus* and *F. oxysporum* was obtained. Films with RE (5%) OO (5%) decreased the number of *P. putida* by 1-log (7.13 × 10^7^ cfu/mL) compared control (2.28 × 10^8^ cfu/mL) and *F. oxysporum* (2.48 × 10^3^ cfu/mL), by 2-log *B. subtilis* (3.35 × 10^5^ cfu/mL), *E. coli* (1.75 × 10^5^ cfu/mL) while control was (3.68 × 10^8^ cfu/mL) and *C. albicans* (1.45 × 10^4^ cfu/mL) compared control samples (7.50 × 10^3^ cfu/mL) and total reduction in *S. aureus* and *A. alternata*. Foil OO (5%) with MP (5%) reduced by 1-1.5-log growth *B. subtilis* (3.35 × 10^5^ cfu/mL), *P. putida* (3.75 × 10^7^ cfu/mL) and *A. alternata* (2.55 × 10^3^ cfu/mL) when the control for this molds was (1.78 × 10^4^ cfu/mL), by 2-log of *C. albicans* (2.06 × 10^4^ cfu/mL) and *F. oxysporum* (3.03 × 10^4^ cfu/mL) and total reduction in *S. aureus*. Next sample RE (2.5%) with A121 (2.5%) reduced only *F. oxysporum* by 1-log (1.46 × 10^4^ cfu/mL) and RE (5%) with A121 (5%) reduced by 1-log *P. putida* (3.88 × 10^7^ cfu/mL). Foil OO (5%) with CO (5%) had total reduction in all bacteria strains and *C. albicans, A. alternata* and *F. oxysporum*, 1-log reduction number of *Aspergillus brasiliensis* (1.75 × 10^2^ cfu/mL) compared to control (3.78 × 10^3^ cfu/mL). Films GTE (5%) with CO (5%) reduced by 1-log number of *B. subtilis* (1.33 × 10^6^ cfu/mL), *S. aureus* (4.95 × 10^4^ cfu/mL) compared to control (6.83 × 10^5^ cfu/mL) and total reduction in *P. putida*. Whereas foil GTE (5%) with A121 (5%) reduced *B. subtilis* by 1-log (6.25 × 10^6^ cfu/mL) and had total reduction in *S. aureus* and *P. putida*. Foil GTE (5%) with OO (%) also reduced *B. subtilis* by 1-log (1.43 × 10^6^ cfu/mL), by 2-log *F. oxysporum* (6.75 × 10^2^ cfu/mL) and had total reduction in *S. aureus, P. putida, C. albicans* and *A. alternata*.

Results obtained for 6 different films with mixtures active substances are presented in Appendix A for bacteria and in Appendix A for yeast and molds. First film containing OO (5%), RE (2.5%), MP (2.5%) reduced number by 1-log *A. alternata* (6.5 × 10^2^ cfu/mL) when the control was (1.78 × 10^4^ cfu/mL), *F. oxysporum* (8.25 × 10^4^ cfu/mL) compared to control (5.47 × 10^5^ cfu/mL), by 2-log *P. putida* (2.77 × 10^6^ cfu/mL) compared to control (2.28 × 10^8^ cfu/mL) and total number of *S. aureus* and *C. albicans*. Foil OO (5%), RE (2.5%) CO (1.75%) reduced number by 1-log of *A. brasiliensis* (5.5 × 10^2^ cfu/mL) while the control was (5.21 × 10^3^ cfu/mL), *Penicilium expansum* (9.25 × 10^3^ cfu/mL) compared to control (1.97 × 10^4^ cfu/mL), more than 2-log reduction number of *B. subtilis* (3.10 × 10^4^ cfu/mL) when the control was (1.47 × 10^7^ cfu/mL) *P. putida* (5.77 × 10^6^ cfu/mL) and total reduction in *E. coli*, *S. aureus, C. albicans, A. alternata* and *F. oxysporum*. Film MP (2.5%), OO (5%), GTE (2.5%) reduced number by 1-log *A. alternata* (6.5 × 10^2^ cfu/mL), *F. oxysporum* (8.25 × 10^4^ cfu/mL), by 2-log *P. putida* and total number of *S. aureus* and *C. albicans*. Foil GTE (2.5%), MP (2.5%), OO (5%), RE (2.5%) reduced number by 1-log of *B. subtilis* (1.43 × 10^6^ cfu/mL), by 2-log *F. oxysporum* (6.75 × 10^4^ cfu/mL) and total number of *S. aureus, P. putida, C. albicans* and *A. alternata*. Next sample MP (1%), OO (5%), RE (2.5%), CO (1.75%) reduced number by 1-log of *B. subtilis* (1.12 × 10^6^ cfu/mL) and total number of *E. coli, S. aureus, P. putida*, *C. albicans, A. alternata* and *F. oxysporum*. Last film GTE (2.5%), MP (1%), OO (5%), RE (2.5%) CO (1.75%) reduced number by 1-log of *B. subtilis* (1.12 × 10^6^ cfu/mL) and total number of *S. aureus, P. putida*, *C. albicans, A. alternata* and *F. oxysporum.* Similar results are described in the literature [38,39,40]. The differences between the numbers of viable cells were not significant, as later confirmed by Duncan’s test (*p* > 0.05). Figure 3, Figure 4, Figure 5 and Figure 6 show the effect of active substances in the extruded films on different strains of bacteria, yeast and molds where there reduced number in growth in the largest number of films tested. A mixture of at least two active agents and with additives of a plasticizer works synergistically and enhances the release of the active substance from the film matrix. This contributes to a total reduction in the growth of tested microorganisms.

The obtained PP films have good hot-sealing properties (confirmed by the similar strength properties of the seal area compared to the non-sealed films—Table 4). The additive presence affect mechanical properties; Young’s modulus was at the level confirming the sealing of the obtained material. The mechanical properties of the sealed parts are closely related to the amount of plasticizing additives introduced into the polymer matrix. The higher the amount of plasticizer, the weaker the sealing strength. Most likely, it can be caused by the phase separation process and higher rate of migration of plasticizers with bioactive agents onto the film surface. This phenomenon (desirable due to the biocidal properties of the film) disturbs the sealing, creating the "easy peel" effect in the sealed places. Ruiz-Cabello and co-authors observed that films containing oregano oil were more flexible, highlighting the plasticization effect of the natural extracts [41].

For the confirmation of the possibility packages formation from our new materials, the prototypes of perforated bags with tomatoes inside were prepared and tested (Figure 7). We compared control PP bags with MP, OO, RE, CO exhibiting the biggest antimicrobial properties. We observed that tomatoes in control had sign of molds, but PP with active substances was still fresh without changes in the appearance. The total number of microorganisms after 14 days storage in control bag PP was 2 × 10^4^ cfu/mL, but for bags with MP, OO, RE, CO was 1.45 × 10^1^ cfu/mL. It shown that releasing of active substances reduced growth microorganism on tomatoes surface.

Our team will continue storage test in the future. Ramos and co-authors suggest that carvacrol show a potential use as antioxidants for active packaging to extend the shelf life of food products [42]. This effect was obtained by filling the film with a biocidal compound relatively high, which directly affects the bioactivity time of the film. In the future, this will increase the production costs of the film, but we believe that storage time of vegetables and fruits packed this way will be prolonged and thus the extended shelf life, will compensate increased packaging production costs.

## 4. Conclusions

This paper presents novel polypropylene-based films modified with active agents and plasticizers with bioactive properties. In order to demonstrate their influence on the functional properties of the film and the antimicrobial potential, films modified with mono, double and mixtures of selected additives were obtained.

The best four compositions of additives were selected; RE with OO, OO with OC, OO, RE, CO and MP, OO, RE, CO acting as both active agents and plasticizers. Thanks to the gradual release from the polymer material, the biocidal activity was enhanced. Films with only bioactive agent exhibited lower or no biocidal properties. In general cases, only higher concentrations of active substances were more effective through microbials. This allowed for the reduction in bioactive additives while extending the biocidal effect. Among all additives presented in this study, oregano oil was the most bioactive. Moreover, OO with cedar oil showed the highest level of protection against microorganisms amongst studied materials, and thus potentially prevents food spoilage. Initial attempts to obtain a sealable bag from new materials have shown their weaker mechanical properties compared to unmodified PP foil. This is related to the rate of “exudation” of the plasticizer and can be easily tunable by its concentration. The sealed areas obtained in this method are characterized by the so-called easy-peel effect, which may be an interesting feature for some film applications.

PP films extruded with double and multi active substances have a wide scope pf activity against bacteria, yeast and molds, which are typical microflora responsible for the spoilage of fresh tomatoes. Storage tests have shown that packages with active substances inhibit the growth of microorganisms on the tomatoes’ surface. They could be potentially applied in the packaging industry as food direct-contact materials with the ability to extend the freshness of selected fresh-market products.

The increased biocidal effect in the case of some PP films modified with special processing additives may be caused by faster migration of the active agent with the plasticizers’ presence. This phenomenon affects higher biocidal concentration of active agents in a shorter time on the PP film surface.

PP films with biocidal properties presented in this work show a great deal of application potential. In the future, they will certainly become an inspiration for the synergy of plasticizers and biocidal substances, and for the development of other types of polymers, especially based on biodegradable materials, e.g., PLA or PHA.

## Figures and Tables

**Figure 1 polymers-13-03478-f001:**
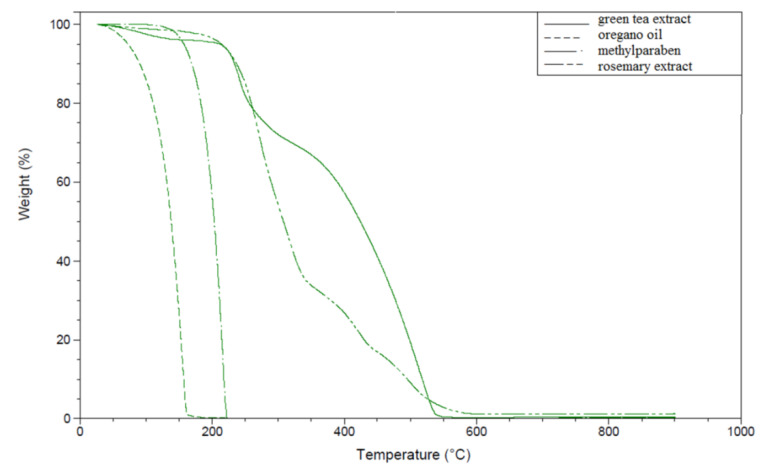
TG curves of selected active agents.

**Figure 2 polymers-13-03478-f002:**
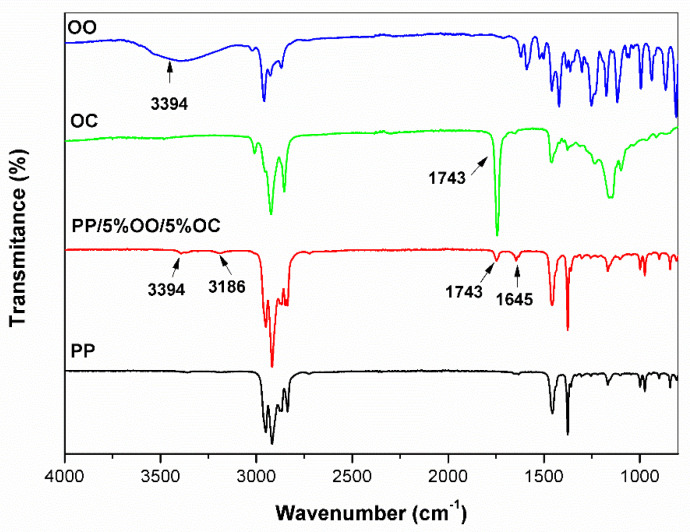
FTIR spectra for Compared spectrum selected additives (OO and CO), PP and modified PP film.

**Figure 3 polymers-13-03478-f003:**
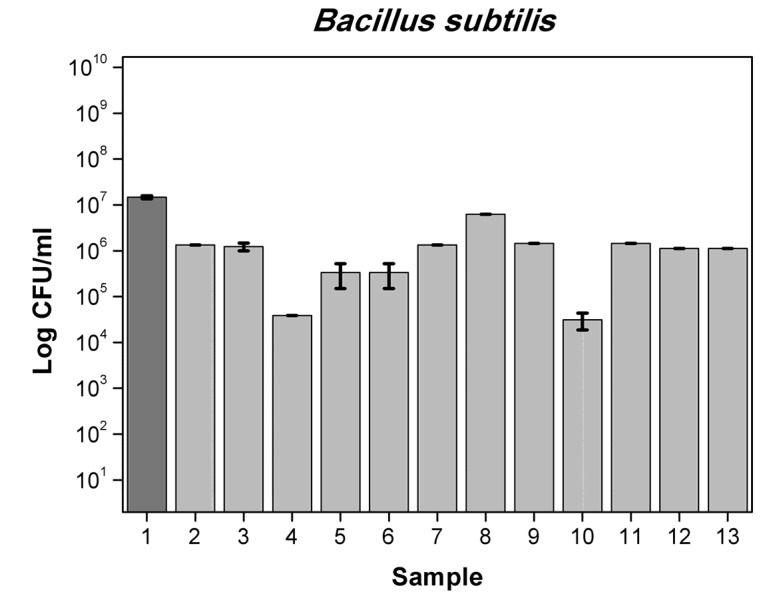
The influence of active substance on *B. subtilis* growth: 1-control, 2-GTE, 3-OO 7.5%, 4-RE,CO, 5-RE,OO, 6-OO,MP, 7-GTE,CO, 8-GTE,A121, 9-GTE,OO, 10-GTE,OO,RE,CO, 11-GTE,MP,OO,RE, 12-MP,OO,RE,CO, 13-GTE,MP,OO,RE,CO.

**Figure 4 polymers-13-03478-f004:**
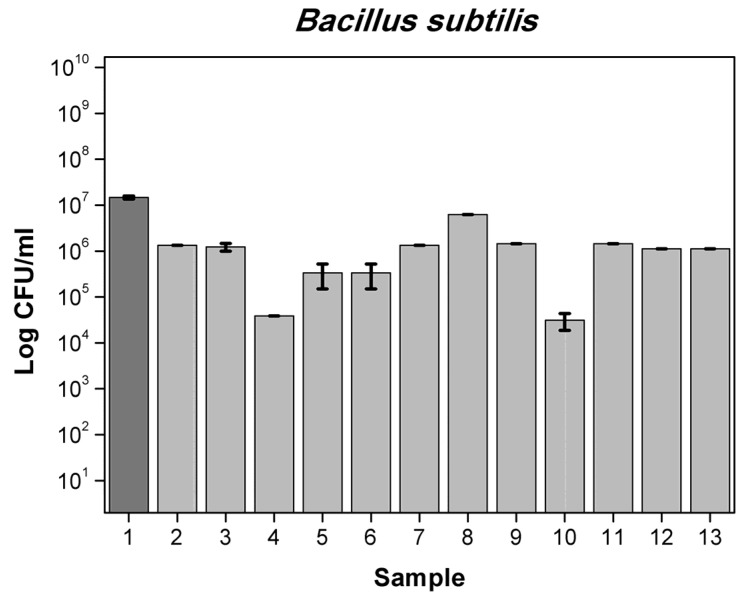
The influence of active substance on *P. putida* growth: 1-control, 2-OO 7.5%, 3-RE,OO, 4-OO,MP, 5-RE,A121, 6-OO,RE,CO, 7-MP,OO,RE.

**Figure 5 polymers-13-03478-f005:**
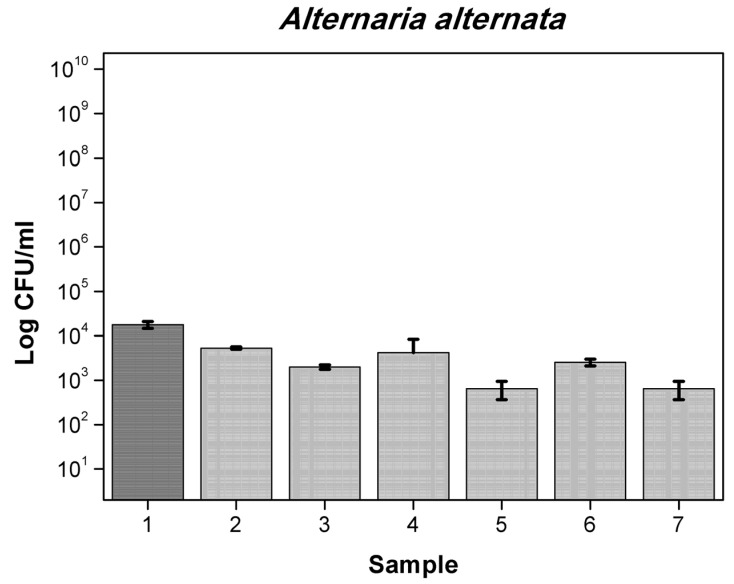
The influence of active substance on *A. alternata* growth: 1-control, 2-CO, 3-RE, 4-OO 5%, 5-OO 7.5%, 6-OO,MP, 7-OO,RE,MP.

**Figure 6 polymers-13-03478-f006:**
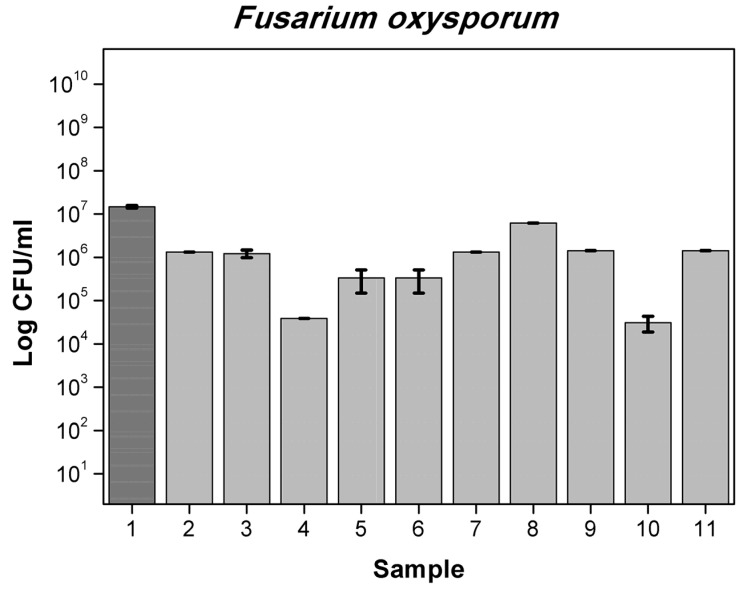
The influence of active substance on *F. oxysporum* growth: 1-control, 2-RE, 3-OO 5%, 4-OO 7.5%, 5-RE,CO, 6-RE,OO, 7-OO,MP, 8-RE,A121, 9-GTE,OO, 10-OO,RE,MP, 11-GTE,MP,OO,RE.

**Figure 7 polymers-13-03478-f007:**
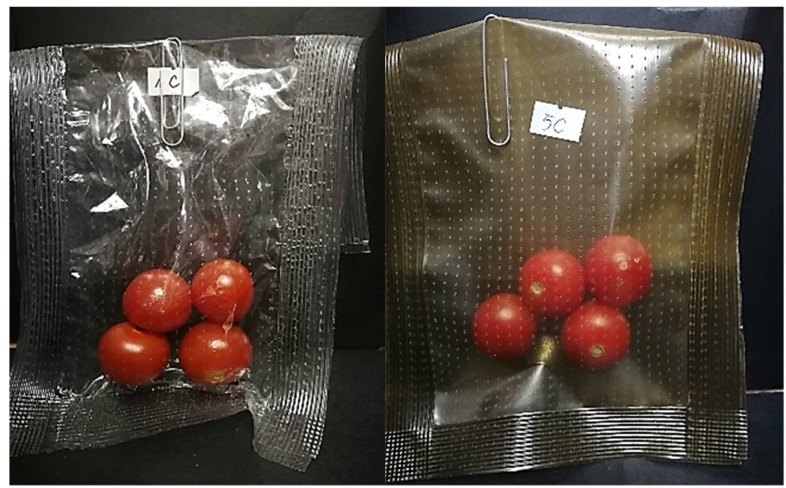
Examples of tomatoes in bag.

**Table 1 polymers-13-03478-t001:** The influence of single additive on the mechanical properties of various PP films.

Active Substance		Young’s Modulus	Tensile Strength	Stress at Break	Elongation at Break	MFI
** *Acronym* **	** *(%)* **	** *(MPa)* **	** *(MPa)* **	** *(MPa)* **	** *(%)* **	** *(g/10 min)* **
w. a. *	-	400 ± 2.1	22 ± 0.9	18 ± 0.7	645 ± 3.8	9.000
A121	5	255 ± 5.2	20 ± 1.2	16 ± 2.0	762 ± 8.2	12.87
CO	5	368 ± 4.2	21 ± 1.1	20 ± 1.1	648 ± 6.8	12.24
GTE	5	676 ± 8.1	29 ± 0.2	27 ± 0.1	780 ± 9.4	7.56
RE	5	397 ± 6.3	21 ± 0.7	19 ± 2.1	662 ± 4.5	8.88
OO	5	319 ± 2.2	22 ± 0.8	18 ± 0.9	665 ± 3.6	11.25
	7.5	366 ± 2.8	26 ± 1.4	21 ± 1.3	636 ± 6.7	13.44
	10	555 ± 3.6	30 ± 1.7	25 ± 1.6	623 ± 4.8	21.20

* w. a.—without additives.

**Table 2 polymers-13-03478-t002:** The influence of double additives on the mechanical properties of PP film.

Active Substance		Young’s Modulus	Tensile Strength	Stress at Break	Elongation at Break	MFI
** *Acronym* **	** *(%)* **	** *(MPa)* **	** *(MPa)* **	** *(MPa)* **	** *(%)* **	** *(g/10 min)* **
w. a *	-	400 ± 2.1	22 ± 0.9	18 ± 0.7	645 ± 3.8	9.00
RE	2.5	398 ± 3.2	22 ± 1.8	22 ± 1.8	628 ± 5.1	12.24
CO	5.0					
RE	5	564 ± 3.9	33 ± 1.6	33 ± 1.6	777 ± 4.5	10.92
OO	5					
OO	5	746 ± 6.1	37 ± 2.9	37 ± 2.9	940 ± 5.8	10.68
MP	5					
RE	2.5	700 ± 8.3	35 ± 1.7	35 ± 1.7	178 ± 7.9	10.32
A121	2.5					
RE	5	790 ± 10.0	31 ± 2.5	31 ± 2.5	986 ± 11.3	11.76
A121	5					
OO	5	432 ± 4.2	31 ± 4.1	30 ± 4.3	866 ± 06.7	15.72
CO	5					
GTE	5	569 ± 7.7	25 ± 2.0	19 ± 1.3	849 ± 9.4	9.72
CO	5					
GTE	5	491 ± 3.5	27 ± 2.3	23 ± 02.1	906 ± 7.3	9.12
A121	5					
GTE	5	360 ± 4.1	20 ± 0.9	16 ± 1.0	689 ± 6.2	9.84
OO	5					

* w. a.—without additives.

**Table 3 polymers-13-03478-t003:** The effect of multi additives mixtures on the mechanical properties of polypropylene (PP) films.

Active Substance		Young’s Modulus	Tensile Strength	Stress at Break	Elongation at Break	MFI
** *Acronym* **	** *(%)* **	** *(MPa)* **	** *(MPa)* **	** *(MPa)* **	** *(%)* **	** *(g/10 min)* **
w. a *	-	400 ± 2.1	22 ± 0.9	18 ± 0.7	645 ± 3.8	9.00
OO	5	582 ± 5.4	30 ± 2.9	30 ± 2.0	841 ± 6.7	12.72
RE	2.5					
MP	2.5					
OO	5	791 ± 7.2	41 ± 1.3	41 ± 1.3	1137 ± 11.9	11.52
GTE	2.5					
CO	1.75					
MP	2.5	521 ± 3.4	30 ± 2.6	38 ± 2.4	936 ± 8.2	9.60
OO	5					
RE	2.5					
GTE	2.5	655 ± 6.3	30 ± 3.8	23 ± 1.7	964 ± 5.7	10.68
MP	2.5					
OO	5					
RE	2.5					
MP	1	488 ± 3.2	30 ± 2.9	27 ± 2.5	978 ± 8.1	11.04
OO	5					
RE	2.5					
CO	1.75					
GTE	2.5	574 ± 5.5	30 ± 1.1	24 ± 3.7	929 ± 7.0	11.74
MP	1					
OO	5					
RE	2.5					
CO	1.75					

* w. a.—without additives.

**Table 4 polymers-13-03478-t004:** Influence of multi additives mixtures on the hot-sealability of polypropylene (PP) films.

Active Substance		Young’s Modulus	Tensile Strength	Stress at Break	Elongation at Break
** *Acronym* **	** *(%)* **	** *(MPa)* **	** *(MPa)* **	** *(MPa)* **	** *(%)* **
w. a *	-	429 ± 3.9	16 ± 0.82	12 ± 0.42	25 ± 1.14
RE	5	100 ± 1.5	2 ± 0.12	4 ± 0.53	2 ± 0.21
OO	5				
OO	5	18 ± 0.9	1 ± 0.11	1 ± 0.07	85 ± 1.96
CO	5				
RE	2.5	88 ± 3.3	2 ± 0.09	4 ± 0.18	4 ± 0.12
OO	5				
CO	1.75				
RE	2,5	86 ± 2.6	5 ± 0.31	10 ± 0.11	23 ± 0.56
OO	5				
CO	1.75				
MP	1				

* w. a.—without additives.

## Data Availability

The data presented in this study are available on request from the corresponding author.

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
