# Peer review of "Development and Characterization of Bioactive Polypropylene Films for Food Packaging Applications"

_polymers, 2021, doi:10.3390/polym13203478_

Round 1
Reviewer 1 Report
Dear Author, I reviewed the revised version of the manuscript (polymers-1326450) entitled: Development and characterization of bioactive poly-propylene films for food packaging applications. This version of the manuscript followed all the suggested modifications and recommendations by the reviewers. Besides, the findings obtained in this research are well described and compared with bibliographical references and justify the importance of this obtained data. For this reason, I considered that this manuscript can be accepted for its publication in this journal.
Author Response
The authors would like to thank to the Reviewer.
With regards
Adrian Krzysztof Antosik
Urszula Kowalska
Magdalena Stobińska
Paulina Dzięcioł
Magdalena Pieczykolan
Katarzyna Kozłowska
Artur Bartkowiak
Reviewer 2 Report
The work does not have the value of a scientific work. This is a technology report. The authors propose to mix polypropylene with a number of modifiers, but there is no consistent scientific hypothesis (!). It is a screening and random approach. The authors did not demonstrate the advisability of mixing bioactive additives. The section "Selection of final antimicrobial substance" is very poorly described. The authors do not provide credible justification as to why they are researching such additions. A poorly described part of the literature in this regard.
In addition, the following detailed notes for work:
- On page 1, all the authors are from the same institution so I don't see any sense in duplicating its name and numbering with the names
- The work is very carelessly formatted, which proves the lack of diligence in writing. 2a. The POLYMERS logo is on page 17.
- Different font size on page 5.
- The introduction part is very weak, it needs to be expanded with the current literature on selected bioactive compounds
- The introduction of compounds whose decomposition / distillation is below processing temperature is a misunderstanding. The authors note this fact but do not try to verify what is happening with this substance.
- No analytics on how much add-on actually was introduced
- There is no analytics or the additive has not oxidized under processing conditions
- No IR spectra additives or composites obtained, no spectroscopic discussion for the product.
- No identification of thermal decomposition products for additives.
- There are no studies to what extent the additives are permanently immobilized in the polymer matrix
- The quality of figure 1 is unsatisfactory
- Figure 1 is irrelevant in the context of scientific work. this should be from the description experimental.
- Tables 1, 2, 3, 4 are 3 pages (!!!) without any significant information. These data should cover no more than half a page of the manuscript, the authors artificially inflate the volume of the manuscript.
- Tables 8 and 9 are too extensive, the authors increase the volume of the manuscript in a negligible way, the same is true for Tables 10, 11, 12, 13
- The work contains a lot of insignificant information in the tables that are appropriate for a technical report, not a scientific publication!
Author Response
The authors would like to thank to the Reviewer and truly appreciate his comments, questions and corrections. Please find the detailed answers below.
The work does not have the value of a scientific work. This is a technology report. The authors propose to mix polypropylene with a number of modifiers, but there is no consistent scientific hypothesis (!). It is a screening and random approach. The authors did not demonstrate the advisability of mixing bioactive additives. The section "Selection of final antimicrobial substance" is very poorly described. The authors do not provide credible justification as to why they are researching such additions. A poorly described part of the literature in this regard.
In addition, the following detailed notes for work:
1. On page 1, all the authors are from the same institution so I don't see any sense in duplicating its name and numbering with the names.
We have made corrections in the article, and we hope it's all right.
2. The work is very carelessly formatted, which proves the lack of diligence in writing. 2a. The POLYMERS logo is on page 17.
We have made corrections in the article, and we hope it's all right.
3. Different font size on page 5.
We have made corrections in the article, and we hope it's all right.
4. The introduction part is very weak, it needs to be expanded with the current literature on selected bioactive compound.
We have made corrections in the article, and we hope it's all right.
5. The introduction of compounds whose decomposition / distillation is below processing temperature is a misunderstanding. The authors note this fact but do not try to verify what is happening with this substance.
During the experiment, we were predicted a partial loss of active substances (especially oregano oil), so we used their higher concentrations, which are antimicrobial effective.
6. No analytics on how much add-on actually was introduced.
Attempts were made to determine the amount of bioactive compounds in the obtained modified films using UV-VIS method. However, it was not possible to determine the amount of substance released at specific time intervals. At the moment, we analyzed the release (over time) of the active substances from the polymer matrix by UV-VIS.
7. There is no analytics or the additive has not oxidized under processing conditions.
Based on the available literature, we assumed that with a content of 5-10% oregano oil, there is no oxidation reaction. [Llana-Ruiz-Cabello M, Pichardo S, Bermúdez JM, Baños A, Núñez C, Guillamón E, Aucejo S, Cameán AM. Development of PLA films containing oregano essential oil (Origanum vulgare L. virens) intended for use in food packaging. Food Addit Contam Part A Chem Anal Control Expo Risk Assess. 2016 Aug;33(8):1374-86]
8. No IR spectra additives or composites obtained, no spectroscopic discussion for the product.
Thank you so much for your suggestion. The analyzed samples were measured by FTIR spectroscopy. We found spectral characteristics between active substance (oregano oil, methyl paraben, rosemary extract, green tea extract) and reference compound (PP) shows that the major ingredients occurring e. g. essential oils dominate the resulting in vibration spectra. Therefore, the spectras of the active substances exhibit molecular vibration profiles very similar to that of their main components.
9. No identification of thermal decomposition products for additives.
The decomposition products of phenolic compounds are: CO, CO2, CH3, H2O, smaller quantities of H2 and formaldehyde. We didn’t observe decomposition of carvacrol during the technological process, because active films would not exhibit antimicrobial properties. Microbiological test have shown that active substances in the processing temperatures don’t change their antimicrobial properties.
10. There are no studies to what extent the additives are permanently immobilized in the polymer matrix
Permanent immobilized of compounds in the matrix was not the objective of this research. Microbiological activity of polypropylene films is based on slow release of biocidal compounds from the polymer matrix. This results in saturation of the substances in a package and reduction of microorganisms. Amount of the active substance is time-varying and tends towards the complete release of compounds from the matrix. In this study, the amount of additives introduced to achieve biocidal effects over time was determined.
11. The quality of figure 1 is unsatisfactory
We have made corrections in the article and quality of figure 1 is better.
12. Figure 1 is irrelevant in the context of scientific work. this should be from the description experimental.
We agree with the reviewer's opinion. However, other reviewers asked for a diagram of the experiment.
13. Tables 1, 2, 3, 4 are 3 pages (!!!) without any significant information. These data should cover no more than half a page of the manuscript, the authors artificially inflate the volume of the manuscript.
We have made corrections. Tables 1-4 have been moved to the Supplementary Material.
14. Tables 8 and 9 are too extensive, the authors increase the volume of the manuscript in a negligible way, the same is true for Tables 10, 11, 12, 13.
We agree with the Reviewer that the tabular form of presentation of the results is not perfect, but the data presented in other forms (e. g. graph) were illegible and misleading.
15. The work contains a lot of insignificant information in the tables that are appropriate for a technical report, not a scientific publication! 
We understand the Reviewer's opinion, and we hope that with the corrections made according to his and other Reviewers' suggestions, the article meets expectations, e. g. Tables 1-4 have been moved to Supplementary Materials.
Finally, we hope that corrections made in the manuscript fulfill reviewer suggestions and allow editor to make positive decision about acceptation of our contribution for publishing in this journal.
With regards,
Adrian Krzysztof Antosik
Urszula Kowalska
Magdalena Stobińska
Paulina Dzięcioł
Magdalena Pieczykolan
Katarzyna Kozłowska
Artur Bartkowiak
Reviewer 3 Report
In this paper, the authors prepared bioactive polypropylene (PP) films with active compounds (ac) presence for food packaging application. Two types of plasticizers (natural compound as well as commercial synthetic product) were investigated. The films physical properties were evaluated. The antimicrobial and biostatic properties of the developed films were also investigated. All materials and methods are described. The results of this paper are good presented. Based on these, this manuscript can be published in “Polymers”, if the authors make the followed changes:
- Page 10, line 336, at Table 6, write the text at the same format
- Page 11, line 345, modify “(Tab.8 - 9)” with “(Tab. 8 - 9)”
- Grammatical mistakes are identifying when the antimicrobial properties are comment. Please, correct: line 379, 381, 382, 389, 395, 420
- Please verify if the values presented at lines 478 and 478 are correctly written.
- Correct the References using the Guide of the Journal.
Author Response
The authors would like to thank to the Reviewer and truly appreciate his comments, questions and corrections. Please find the detailed answers below.
In this paper, the authors prepared bioactive polypropylene (PP) films with active compounds (ac) presence for food packaging application. Two types of plasticizers (natural compound as well as commercial synthetic product) were investigated. The films physical properties were evaluated. The antimicrobial and biostatic properties of the developed films were also investigated. All materials and methods are described. The results of this paper are good presented. Based on these, this manuscript can be published in “Polymers”, if the authors make the followed changes:
- Page 10, line 336, at Table 6, write the text at the same format
We have made corrections in the article, and we hope it's all right.
- Page 11, line 345, modify “(Tab.8 - 9)” with “(Tab. 8 - 9)”
We have made corrections in the article, and we hope it's all right.
- Grammatical mistakes are identifying when the antimicrobial properties are comment. Please, correct: line 379, 381, 382, 389, 395, 420
We have made corrections in the article, and we hope it's all right.
- Please verify if the values presented at lines 478 and 478 are correctly written.
We have made corrections in the article, and we hope it's all right.
- Correct the References using the Guide of the Journal.
We have made corrections in the article, and we hope it's all right.
Finally, we hope that corrections made in the manuscript fulfill reviewer suggestions and allow editor to make positive decision about acceptation of our contribution for publishing in this journal.
With regards
Adrian Krzysztof Antosik
Urszula Kowalska
Magdalena Stobińska
Paulina Dzięcioł
Magdalena Pieczykolan
Katarzyna Kozłowska
Artur Bartkowiak
This manuscript is a resubmission of an earlier submission. The following is a list of the peer review reports and author responses from that submission.
Round 1
Reviewer 1 Report
The work is interesting and valuable. However, it is worth correcting or supplementing certain issues.
Tab. 2. Parameters of the PP re-granulation processor and Tab. 3. Cast processing parameters of modified PP film extrusion can be transferred to supplementary materials
the temperature of the beginning of distillation of oil from oregano is slightly above 100 degrees (Fig. 1), in this case, a discussion should be made about the possibility of loss of this additive during the processing process and the possible possibility of its decomposition.
It is suggested to perform the FTIR test in order to determine the stability of the modifier after the processing process and its possible decomposition. It would be advisable to discuss possible by-products of additive decomposition under processing conditions and their safety for the final application.
Table 4 is suggested to be rebuilt which is very extensive for the information it contributes.
The content of bioactive compounds is very high, from a practical point of view it can be very problematic (relation of the cost of the additive to the cost of packaging) - a discussion is necessary.
Probably an error in the e-mail address : .Antosik@zut.edu.pl
Author Response
Response to the Reviewer
The authors would like to thank to the Reviewer and truly appreciate his comments, questions and corrections. Please find the detailed answers below.
The work is interesting and valuable. However, it is worth correcting or supplementing certain issues.
Tab. 2. Parameters of the PP re-granulation processor and Tab. 3. Cast processing parameters of modified PP film extrusion can be transferred to supplementary materials
Conclusions corrected according to Reviewer’s suggestion.
the temperature of the beginning of distillation of oil from oregano is slightly above 100 degrees (Fig. 1), in this case, a discussion should be made about the possibility of loss of this additive during the processing process and the possible possibility of its decomposition.
We have made corrections in the article, and we hope it's all right.
It is suggested to perform the FTIR test in order to determine the stability of the modifier after the processing process and its possible decomposition. It would be advisable to discuss possible by-products of additive decomposition under processing conditions and their safety for the final application.
Thank you so much for your suggestion. We observed that oregano oil to extend from the polymer matrix – confirm this antimicrobial properties. Therefore, we considered this sufficient confirmation. The intensity of the peaks on FTIR is quite low, it is related to the amount of additives at the test place, therefore we have not introduced the spectrum to publication (sample figure is below).
Table 4 is suggested to be rebuilt which is very extensive for the information it contributes.
Conclusions corrected according to Reviewer’s suggestion.
The content of bioactive compounds is very high, from a practical point of view it can be very problematic (relation of the cost of the additive to the cost of packaging) - a discussion is necessary.
We have made corrections in the article, and we hope it's all right.
Probably an error in the e-mail address : .Antosik@zut.edu.pl
We have made corrections in the article.
Finally, we hope that corrections made in the manuscript fulfill reviewer suggestions and allow editor to make positive decision about acceptation of our contribution for publishing in this journal.
With regards
Adrian Krzysztof Antosik
Urszula Kowalska
Magdalena Stobińska
Paulina Dzięcioł
Magdalena Pieczykolan
Katarzyna Kozłowska
Artur Bartkowiak

Reviewer 2 Report
Dear Author, I reviewed the manuscript (polymers-1224342) entitled Blends of polyolefin films with bioactive compounds. This manuscript presents relevant information about the potential application of polyolefin films with bioactive compounds. However, some sections of the presented data can be improved. For this reason, I considered that this manuscript needs minor changes for being considered for its publication in this journal.
Additional comments.
Highlight the advantages of using these films with bioactive compounds to preserve food safety.
Check paragraphs extension in this manuscript.
Check typo in the microbial names section.
Try to include a bibliographical reference in the antibacterial assay protocol performed in this manuscript.
Include an experimental design that contents statistical factors and variables of response in the statistical analyses applied to the findings of this research.
Include a possible mode of bioactive compounds against pathogenic bacteria in the tested assays.
Try to compare the obtained findings with similar assays where films with bioactive compounds were applied to preserve food safety.
Try to include a statistical description in the figures that required it.
Include future trends to keep working with the obtained data.
Try to conclude with a general statement of the most relevant part of this study.
Author Response
Response to the Reviewer
The authors would like to thank to the Reviewer and truly appreciate his comments, questions and corrections. Please find the detailed answers below.
Dear Author, I reviewed the manuscript (polymers-1224342) entitled Blends of polyolefin films with bioactive compounds. This manuscript presents relevant information about the potential application of polyolefin films with bioactive compounds. However, some sections of the presented data can be improved. For this reason, I considered that this manuscript needs minor changes for being considered for its publication in this journal.
Additional comments.
Highlight the advantages of using these films with bioactive compounds to preserve food safety.
We have made corrections in the article, and we hope it's all right.
Check paragraphs extension in this manuscript.
We have made corrections in the article, and we hope it's all right.
Check typo in the microbial names section.
We have made corrections in the article, and we hope it's all right.
Try to include a bibliographical reference in the antibacterial assay protocol performed in this manuscript.
We have complement, and we hope it's all right.
Include an experimental design that contents statistical factors and variables of response in the statistical analyses applied to the findings of this research.
We have complement, and we hope it's all right.
Include a possible mode of bioactive compounds against pathogenic bacteria in the tested assays.
We have wrote in article, and we hope it's all right.
Try to compare the obtained findings with similar assays where films with bioactive compounds were applied to preserve food safety.
We have wrote in article, and we hope it’s all right.
Try to include a statistical description in the figures that required it.
We have made corrections in the article, and we hope it's all right.
Include future trends to keep working with the obtained data.
We have wrote in article, and we hope it’s all right.
Try to conclude with a general statement of the most relevant part of this study.
We have wrote in article, and we hope it’s all right.
Finally, we hope that corrections made in the manuscript fulfill reviewer suggestions and allow editor to make positive decision about acceptation of our contribution for publishing in this journal.
With regards
Adrian Krzysztof Antosik
Urszula Kowalska
Magdalena Stobińska
Paulina Dzięcioł
Magdalena Pieczykolan
Katarzyna Kozłowska
Artur Bartkowiak
Reviewer 3 Report
The document blends of polyolefin films with bioactive compounds is a manuscripts that describes some experiments performed in characterization and antimicrobial activities evaluation on polyolefin films after the addition of bioactive compounds. There are a lot of work in the literature that evaluate in a deeper and interesting way the addition of bioactive compounds in this kind of films, so this document lacks of novelty and analysis.
Here you will find some general comments.
In materials and methods section the table 1 could be deleted and the acronyms of the compounds must be expressed in the text.
Table 2 is not clearly explained, please explain “zone” in the text or in table description
The statistical analysis is not mentioned
In Results and discussion section, in line 171 please explain the value of “relatively high temperature”. A better explanation and discussion of the behavior of the TG curves of active compounds must be added.
The treatments of modifications of PP film by plasticizers and bioactive compounds must be explained by a specific experimental design, and the resulted variable or variables must be clearly explained.
It is not clear why all the standard deviations are 0.1 in table 5, significantly differences are not presented among the values. A better discussion and analysis of the data must be carried out.
For the results of antimicrobial effect the effect of the bioactive compounds directly as a control must be added, or at least the MIC’s.
In general the information of antimicrobial activity is poorly detected, and it is possible that the methodologies used or the concentrations applied are not adequate.
Table 8 IDEM with table 5
Antimicrobial activity of mixtures (table 10) follows the same behavior of the PP films with one bioactive compound.
Figure 4 and 5 are not clearly presented. In the case of the mixtures of bioactive compounds that presented total reduction a better explanation of a possible additive or synergic effect must be added.
The antimicrobial effect on tomato samples is not presented. For the last paragraph from 359- 364 there is not enough evidence.
Conclusions are not well supported by the results.
Author Response
Response to the Reviewer
The authors would like to thank to the Reviewer and truly appreciate his comments, questions and corrections. Please find the detailed answers below.
The document blends of polyolefin films with bioactive compounds is a manuscripts that describes some experiments performed in characterization and antimicrobial activities evaluation on polyolefin films after the addition of bioactive compounds. There are a lot of work in the literature that evaluate in a deeper and interesting way the addition of bioactive compounds in this kind of films, so this document lacks of novelty and analysis.
The novelty of the article is the use of the "exudation" effect of the plasticizer from the polymer matrix, which enhances the biocidal effect of the film (films only with bioactive compounds would have lower or no biocidal activity, only much higher fillings would cause a similar biocidal effect). This allowed for a dramatic reduction in bioactive additives while extending the biocidal effect.
Here you will find some general comments.
In materials and methods section the table 1 could be deleted and the acronyms of the compounds must be expressed in the text.
Conclusions corrected according to Reviewer’s suggestion.
Table 2 is not clearly explained, please explain “zone” in the text or in table description
We agree with the reviewer information defining the zones has been added in Tables 2 and 3
The statistical analysis is not mentioned
We have made corrections in the article, and we hope it's all right.
In Results and discussion section, in line 171 please explain the value of “relatively high temperature”.
This temperature is high for biocidal additives, e.g. for oregano oil, which has 100% weight loss at this temperature (TG test). However, due to the production process (extrusion), these specimens are trapped in the polymer mass and evaporate from the surface of the film. By appropriately lowering the extrusion temperature and increasing the cooling rate, losses can be minimized to about 15-25% of the compound feed to the polymer. Conclusions corrected in manuscript.
A better explanation and discussion of the behavior of the TG curves of active compounds must be added.
Conclusions corrected according to Reviewer’s suggestion.
The treatments of modifications of PP film by plasticizers and bioactive compounds must be explained by a specific experimental design, and the resulted variable or variables must be clearly explained.
We have complement in the article, and we hope it's all right.
It is not clear why all the standard deviations are 0.1 in table 5, significantly differences are not presented among the values. A better discussion and analysis of the data must be carried out.
We agree with the reviewer, it was a editing issue. The deviation values entered in the table have been corrected.
For the results of antimicrobial effect the effect of the bioactive compounds directly as a control must be added, or at least the MIC’s.
We have made corrections in the article, and we hope it's all right.
In general the information of antimicrobial activity is poorly detected, and it is possible that the methodologies used or the concentrations applied are not adequate.
This is because small amounts of bioactive compounds are "trapped" in the polymer matrix. As a result, despite the release of the compound from the film, it does not achieve a biocidal concentration on its surface. For this purpose, in order to accelerate the release process of the biocidal compound, synergy with a plasticizer was used, which accelerates the release of bioactive components during the "exudation" process. This effect makes it possible to achieve the level of biocide with lower concentrations of additives. We have made corrections in the article, and we hope it's all right.
We used a standard method ASTM E 2180-01 is required to the antimicrobial agents in polymeric or hydrophobic materials.
Table 8 IDEM with table 5
Tables 8 and 5 are similar but not the same - they show the effect on the mechanical properties of the films of one and two compounds, respectively. They have not been saved as one table for the sake of easier presentation of the results. Based on the results, multi-component compositions included in Table 9 were created.
Antimicrobial activity of mixtures (table 10) follows the same behavior of the PP films with one bioactive compound.
We have made corrections in the article, and we hope it's all right.
Figure 4 and 5 are not clearly presented. In the case of the mixtures of bioactive compounds that presented total reduction a better explanation of a possible additive or synergic effect must be added.
We have made corrections in the article.
The antimicrobial effect on tomato samples is not presented. For the last paragraph from 359- 364 there is not enough evidence.
We added more information about the antimicrobial effect on tomato samples.
Conclusions are not well supported by the results.
We agree with the reviewer, we hope that changes introduced in line with the suggestions and recommendations of the reviewers fixed this issue.
Finally, we hope that corrections made in the manuscript fulfill reviewer suggestions and allow editor to make positive decision about acceptation of our contribution for publishing in this journal.
With regards
Adrian Krzysztof Antosik
Urszula Kowalska
Magdalena Stobińska
Paulina Dzięcioł
Magdalena Pieczykolan
Katarzyna Kozłowska
Artur Bartkowiak
Round 2
Reviewer 1 Report
Please remove the device tag (Universal V4.3 TA ...) from figure 2
The diagram in Figure 1 should be of a better quality, at least 300 dpi
Author Response
The authors would like to thank to the Reviewer and truly appreciate his comments, questions and corrections. Please find the detailed answers below.
Please remove the device tag (Universal V4.3 TA ...) from figure 2
We have made corrections in the article.
The diagram in Figure 1 should be of a better quality, at least 300 dpi
We have made corrections in the article, and we hope it's all right.
Finally, we hope that corrections made in the manuscript fulfill reviewer suggestions and allow editor to make positive decision about acceptation of our contribution for publishing in this journal.
With regards
Adrian Krzysztof Antosik
Urszula Kowalska
Magdalena Stobińska
Paulina Dzięcioł
Magdalena Pieczykolan
Katarzyna Kozłowska
Artur Bartkowiak
Reviewer 3 Report
Authors increased the quality of the document “blends of polyolefin films with bioactive compounds”, nevertheless some issues about the experimental design must be corrected.
Please try to explain the type of the experimental design used (Single-factor designs, multi-factor designs) if this is randomized block…..
When talking about statistical analysis in tables we refer to an ANOVA and statistical comparative analysis such as Tukey, Duncan, LSD, please try to included.
Author Response
The authors would like to thank to the Reviewer and truly appreciate his comments, questions and corrections. Please find the detailed answers below.
Authors increased the quality of the document “blends of polyolefin films with bioactive compounds”, nevertheless some issues about the experimental design must be corrected.
Please try to explain the type of the experimental design used (Single-factor designs, multi-factor designs) if this is randomized block…..
We have made corrections in the article, and we hope it's all right.
When talking about statistical analysis in tables we refer to an ANOVA and statistical comparative analysis such as Tukey, Duncan, LSD, please try to included.
We have made corrections in the article, and we hope it's all right.
Finally, we hope that corrections made in the manuscript fulfill reviewer suggestions and allow editor to make positive decision about acceptation of our contribution for publishing in this journal.
With regards
Adrian Krzysztof Antosik
Urszula Kowalska
Magdalena Stobińska
Paulina Dzięcioł
Magdalena Pieczykolan
Katarzyna Kozłowska
Artur Bartkowiak